# Patients’ Expectations and Preferences for the Organizational Conditions of the Colorectal Cancer Screening Programme in Poland: A Qualitative Analysis

**DOI:** 10.3390/healthcare11030371

**Published:** 2023-01-28

**Authors:** Aleksandra Gac, Katarzyna Joanna Kędzior, Katarzyna Pogorzelczyk, Agnieszka Wojtecka, Małgorzata Wojnarowska, Marlena Robakowska, Olga Kalinowska-Beszczyńska, Maria Libura, Katarzyna Kolasa, Włodzimierz Cezary Włodarczyk, Dominik Dziurda, Roman Topór-Mądry, Łukasz Balwicki

**Affiliations:** 1Department of Public Health and Social Medicine, Medical University of Gdansk, 80-210 Gdansk, Poland; 2Agency for Health Technology Assessment and Tariff System, 00-032 Warsaw, Poland; 3Center for Competence Development, Integrated Care and e-Health, Medical University of Gdansk, 80-210 Gdansk, Poland; 4Department of Didactics and Medical Simulation, University of Warmia-Mazury, 10-718 Olsztyn, Poland; 5Division of Health Economics and Healthcare Management, Kozminski University, 03-301 Warsaw, Poland; 6Institute of Public Health, Jagiellonian University, 30-348 Cracow, Poland

**Keywords:** colorectal cancer, CRC, screening, patient expectations, patient preferences

## Abstract

(1) Background: Colorectal cancer (CRC) is a serious health problem in Poland as well as many European Union countries. The study aimed to describe factors that, from the patient’s perspective, could increase the attendance rate and regularity of participation in the colorectal cancer screening programme (SP); (2) Methods: The qualitative approach was applied. The study involved six focus interviews conducted with 24 respondents (12 women and 12 men) aged 40–49, who had at least one first-degree family member diagnosed with CRC and persons aged 50–65, living in five selected voivodships (provinces) of Poland. The collected data were thematically coded. Further, a comparative analysis was conducted, and aggregated statements were formulated; (3) Results: The inclusion of primary care clinics within the CRC SP organization was reported as a key factor in improving the attendance rate and regularity of patient participation in the programme. Particularly important factors included an invitation in the form of a personal letter or a phone call made by staff from primary care clinics; (4) Conclusions: Patients were confirmed to have clear expectations and preferences for the organizational conditions of the CRC SP. Preferences nature allows them to be treated as one of the potential criteria for selecting critical parameters of CRC SPs.

## 1. Introduction

Colorectal cancer (CRC) and its sequelae are a serious health problem in Poland as well as in many European Union (EU) countries [1]. The estimated CRC mortality rate for 2020 in the EU is 24.5/100,000 women and 43.1/100,000 men, while the estimated incidence rate is 56.3/100,000 women and 91.6/100,000 men [1]. In Poland, CRC is the third most common cancer in both sexes [2]. The mortality rate estimated for 2020 in Poland is 30.5/100,000 women and 63.3/100,000 men, while the incidence rate is 51.2/100,000 women and 99.9/100,000 men [1]. CRC incidence and mortality correlate with shortages in healthcare resources, inequitable access to healthcare services, and differences in participation in the screening programme (SP) [3].

### 1.1. Colorectal Cancer Screening Programme in Poland 

In Poland, as part of the National Cancer Strategy (NCS), a publicly funded CRC SP has been in place since 2000 [4]. The programme is based on performing colonoscopies and includes two types of screening: population-based targeted screening (carried out within a system with personal screening invitation letters) and opportunistic screening. It is possible to take a colonoscopy once every 10 years as part of free screening for CRC. The population-based targeted screening relays on sending out personal one-time letter invitations. Participants are recruited from people between the ages of 55 and 64 (regardless of the presence of clinical symptoms seen in the last few months). The opportunistic screening includes four target groups: (1) people aged 50–65, regardless of family history; (2) people aged 40–49 who have a first-degree relative diagnosed with CRC; (3) people aged 25–49 from a family with Lynch syndrome; (4) people aged 20–49 from a family with familial adenomatous polyposis syndrome. The examination can be performed under local anaesthetic. The qualification for the procedure is made by the doctor performing the examination.

Despite the many improvements in the CRC SP that were implemented by the organizers over the period 2000–2021 [5], the attendance rates are not satisfactory in Poland. In 2018, less than 15% of people who received a personal invitation letter underwent a screening colonoscopy examination. The average attendance rate increased to 17% between 2014 and 2018 [6]. However, still remains lower compared to the range of 23–70% achieved in EU countries [7,8].

To encourage patients to take part in screening tests, a nationwide campaign is implemented within the NCS for 2020–2030, entitled “Planuję długie życie” (“Planning for a long life”) [9]. It includes TV and radio commercials, posters and information brochures available on the website of the programme and in healthcare institutions.

### 1.2. Colorectal Cancer Screening Programmes and Patient Preferences

The effectiveness of the CRC SP is determined by the actual attitudes of patients related to participation in the programme [10]. The task of SP organizers is, therefore, to look for a programme construction that ensures not only access to diagnostic procedures but also their actual use by the target population. In addition to the defined types of diagnostic tests, the defined target group and the method of financing, the design of the SP also takes into account different types of organizational conditions of the programme. These are: ways to reach the target population, promotion channels, conditions regarding personnel performing the examination, conditions at the medical facility, territorial accessibility etc. Indicated conditions may be particularly important to ensure patient attendance in CRC SP, in which colonoscopy is the primary examination. This is mainly due to the invasiveness of the procedure itself and the discomfort it causes the patients [11]. 

The authors of the study intended to increase the knowledge about Polish patients’ expectations and preferences regarding the design of the CRC SP. Thus the study aimed to describe factors that—from the patients’ perspective—could increase the attendance rate and regularity of patient participation for the CRC SP in the future.

## 2. Materials and Methods

### 2.1. Study Background

The study constitutes the first phase of the “Comprehensive model of healthcare service analysis” (KOMPAS) project carried out by the Agency for Health Technology Assessment and Tariff System in Poland (AOTMiT) under the project “Maps of Health Needs—System and Implementation Analysis Database” funded by the EU [12]. 

The ethics approval was granted by the Bioethics Committee of the National Institute of Public Health—National Institute of Hygiene (Opinion No. 8/2020).

### 2.2. Study Design

The first step required the development of the research tool (interview scenario), which was later applied during the data collection process. It was achieved as a result of a workshop organized by the AOTMiT with 6 patient organisations and foundations for CRC patients, as well as a literature review (November 2019). The final interviews were preceded by 2 pilot interviews to test the scenarios for their practical application. The pilot study highlighted a few ambiguities in the questionnaire formulation; thus, the required clarifications were introduced. 

The final shape of the research tools was given by the research team of the section for patient preference studies of the KOMPAS project, consisting of representatives of different disciplines: public health, pharmacy, sociology and health economics. 

The scenario of the interviews included a short introduction explaining what screening tests are and what their purpose is. Prior to the discussion, a further explanation was provided on what colonoscopy is and who in Poland is tasked with the performance of examinations under the publicly funded CRC SP. The questions asked during the interviews referred to the following themes: preferred promotional and educational actions and preferences on the health care institution. The exact questions can be found in Table 1 below. 

### 2.3. Recruitment Criteria and Sample

The sample included participants who fulfilled the formal inclusion criteria in the Polish CRC SP in an opportunistic screening model followed by additional inclusion criteria indicated in detail in Appendix A. (Table A1. Recruitment checklist). The formal inclusion criteria to the CRC SP contained persons fulfilling 2 criteria: firstly, without diagnosed CRC and without symptoms such as gastrointestinal bleeding, diarrhoea, or constipation that has occurred in the past few months and whose cause is unknown, as well as weight loss or anaemia without a known cause; secondly, aged 40–49, who had at least 1 first-degree family member diagnosed with CRC and/or persons aged 50–65. 

The research consisted of 6 focus groups, which included equal numbers of participants in terms of gender, place of residence, voivodship (province) of residence, education, and experience of colonoscopy. The indicated variables constituted the additional inclusion criteria of the study, as mentioned above. 

The respondent panel of the research agency which conducted the study was utilized to identify potential participants. Individuals were contacted via phone and invited to participate in the cancer prevention interview. Simultaneously, the pre-qualifier for the study was conducted (based on the checklist criteria; Appendix A). Recruitment was carried out by a professional from a research agency with many years of experience in conducting qualitative social research. The respondents were compensated for their participation in the study as part of their membership in the research agency’s panel. Participants aware of the membership conditions in the panel knew the form and value of compensation for attending the interview. According to the practice of research agencies, the total of points collected for participation in various surveys on an individual user’s profile allows the user (responder) to choose a bonus (such as a shopping voucher).

Purposive sampling was used to recruit participants in order to fulfil the indicated criteria. Individuals were recruited in 5 out of 16 voivodeships (provinces) in Poland. The study selected voivodeships (provinces) with varying levels of demand for CRC treatment services (Appendix B) (Table A2. Results of the rate of provided hospital services within ICD-10 codes C18-C21 per 100,000 inhabitants of individual voivodships in Poland.) and ensured the representation of geographically diverse regions of Poland. Sampling continued until saturation was achieved. 

The final sample consisted of 24 respondents (12 women and 12 men): 13 persons aged 40–49 (who had at least 1 first-degree family member diagnosed with CRC) and 11 persons aged 50–65, living in 5 selected voivodships (provinces) of Poland. The structure and total sample size were assumed a priori by the researchers and are in line with qualitative research practice. Some of the recruited people (45%) refused to participate in the study (mainly due to lack of time for the interview and unwillingness to participate in the focus study).

### 2.4. Data Collection

Data were collected during focus group interviews (FGI) from 3 August to 4 September 2020. Interviews lasted between 1 and 1.5 h. As the study was implemented during the COVID-19 epidemic, appointments were held online using a group communication platform [13]. The direct implementation of the study was commissioned by a research agency responsible for recruiting respondents, conducting and moderating the FGI and preparing transcripts. The moderators have a sociological background and a minimum of 10 years of experience in using qualitative research methods, moderating a group in FGI research, including health care issues. Respondents’ faces were not visible during the interviews (the moderator could only hear their voices). The introductory statements and questions read out by the moderator were presented on the screen. The moderator’s guidance clearly indicated allowing participants to speak spontaneously. 

### 2.5. Data Analysis

The transcribed interviews were coded independently by 2 researchers. The thematic coding strategy was applied. The coding was run manually and aimed at identifying the main themes arriving from the actual statements delivered by participants. The individual codes referred to the experiences of the current organization of CRC screening and to the responders’ expectations.

Generated codes were analyzed and grouped into categories by a research team. Categories were organized in a hierarchical order, from general to detail. The prepared codes provided the basis for a comparative analysis of respondents’ expectations and preferences towards CRC SP. Differences and similarities inside and between categories were formulated, and general conclusions were drawn. Statements representative of the categories were selected and included in the described results. The team of analysts consisted of 5 people with a minimum of 5 years of experience in social research studies. At all stages, the analysis of respondents’ statements was performed by teams of 2 and 3 people working independently. The degree of consistency between the teams performing the analysis was verified. The resolution of disagreements was made by a consensus method.

## 3. Results

The sociodemographic characteristics of the whole group of participants (24 individuals) are shown in Appendix C (Table A3. Sociodemographic characteristics of participants). Characteristics include information on whether or not attendees have participated in CRC SP in the past. 

The analysis of the focus interview results included the aggregation of respondents’ statements to select interrelated opinions, commonalities, and recurring opinions and to establish a hierarchy (frequency) of expressed expectations. The authors’ interpretations and arguments for their conclusions are documented and supported by quoted responses. The statements were divided according to the subjects identified in Table 1 and described according to the characteristics of the respondents provided in Appendix C.

Considering the specific themes following results were obtained:*1.* *Preference for actions concerning the promotion of preventive examinations and education aimed directly at the patient*

The fundamental question refers to who should assess individual risk factors for CRC, determine whether the patient is eligible to participate (according to the terms of the CRC SP) and inform patients about the possibility of a screening colonoscopy.

In response to this question, respondents indicated their primary care (PC) physician as the key person, mainly because of the belief that he or she has access to their complete medical records.


*“Especially as the GP has some history there, insight and so on.”*
[LU-2]

The respondents argued that a primary care physician knows the patient, his or her past illnesses, medications taken, or general psychophysical health, and because of his or her role, knows where to refer the patient for appropriate and comprehensive diagnostics along with treatment if needed.

Moreover, the respondents also indicate that primary care physicians are highly trusted: 


*“The only person who would be able to get me to do this would be a doctor I trust, which is my GP.”*
[MA-3]

Opinions were divided with regard to the role of the occupational physician (OP) in this issue. Some respondents believe that OPs do not have sufficient knowledge and access to information about the patient and the patient’s medical history, nor do they have experience in ongoing patient care. When recounting their experiences of contact with occupational physicians, respondents pointed out that it is regular contact but of short duration and at large intervals. Statements from respondents indicate that occupational physicians are less trusted than primary care physicians. 


*“Definitely not (...) for fear of losing our jobs, we might withhold some information”.*
[LU-1]

The remaining respondents indicated that this form of information and qualification of patients for screening colonoscopy could include people who do not use health care on a regular basis, as well as those who are not aware that this examination can be publicly funded. 


*“For most Poles, this is the only doctor who examines them regularly.”*
[ŚW-3]

Other medical staff, such as primary care nurses or administrative staff, were not mentioned during the discussion. 

As part of the discussion, the moderator provided the respondents with an overview of the idea of healthcare educators (HCE) in preventive care in Poland. The HCE was defined as a person who would support physicians (PC physicians, OPs) in the implementation of procedures for health promotion and education on healthy lifestyles and preventive examinations to control health. Respondents were informed about the key tasks that an HCE could perform under medical supervision and in collaboration with nurses and clinic administration. These include, among others: conducting a detailed interview to assess an individual’s risk of cancer, including CRC, informing the patient about the possibility of participating in the SP and the conditions of the programme, imparting knowledge on healthy lifestyles and diseases of affluence in a way that is accessible and comprehensible to patients. 

Several people in the surveyed group reacted very positively to the proposal to introduce HCE in the Polish healthcare system. They argued that the creation of such a job position would partly relieve doctors in performing the procedures in question while at the same time providing doctors with an opportunity to devote more time to treatment activities. However, the majority of respondents had a negative view of the concept presented to them. For them, HCEs are unnecessary mainly due to the belief that the tasks in question should be performed by primary care physicians.


*“I think this is too much red tape and needless job posts where such information can just as well be provided by a GP.”*
[DŚ-4]

Respondents stressed that they would not trust HCEs or believe that their medical competence is sufficient even if they were employees of a health care provider. Respondents declared that even if HCE services were more easily accessible, they would still opt for a doctor. The HCE is perceived as a stranger, which could present difficulties in conducting an honest, open interview. 


*“The family doctor (...) first of all knows how to talk, because every patient is different, that a lot of people will not open up to a stranger.”*
[LU-2]

In further discussion, letters, preferably personal, were mentioned as other effective forms of directly reaching and encouraging patients to participate in the CRC SP (in addition to PC physicians, OPs, and HCEs). Respondents also mentioned the telephone call, and it was stressed that it is very important for the call to be made from the health centre they belong to. 


*“Telephone outreach to individual patients is better, but with that said, it would have to be from our clinic, not a nationwide system that could be associated with other ventures.”*
[ZP-2]

Respondents were also asked whether the possibility of a prior telephone consultation or anonymous email/chat with a doctor (without the need to appear at the clinic) would, in their opinion, help decide whether to participate in the screening. The possibility of an earlier teleconsultation with a doctor was considered important by the respondents, with a few respondents indicating that they would prefer to see a doctor in person. The possibility of prior anonymous online correspondence (email or chat) with a doctor was not assessed as necessary or helpful in making a decision. 


*“It’s nothing embarrassing if someone wants to take care of their health, to anticipate some situations that might occur (...) For me, it wouldn’t matter”.*
[LU-2]

Informing patients about the CRC SP using SMS was rated by respondents as a distinctly ineffective way to reach patients and improve attendance for the programme. One respondent stated that although “everyone can read an SMS,” it is not an effective form of promotion, especially for older people. A similar question regarding the use of e-mail was answered negatively by the respondents. They justified their opinion by the excess of unsolicited electronic messages (SPAM) and the potential difficulty in reaching older people.

*2.* 
*Preferences for actions on the promotion of population-based targeted screening*


The studied cohort was asked how the nationwide, effective promotion of screening tests should look to ensure that healthy people at high risk of CRC (i.e., eligible for the CRC SP) report for screening without additional encouragement. The discussion was intended to determine which channels of indirect promotion would be most effective in respondents’ opinions. 

Internet and TV campaigns were indicated as the most effective form (based on the frequency of registration), recognizing the superior role of TV.


*“In our country, television is leading the way, and it seems to me that it should be considered in the first place”.*
[LU-2]

The participants also indicated the potential benefits of involving celebrities, persons of public trust or athletes in the promotion. A “well-known male”, aged ≥ 50 years, was suggested as a credible person for the promotional campaign. Two participants stated during the discussion that this role does not necessarily have to be played by a well-known person, but it is important for this person to be credible. A strongly negative opinion was expressed about the possible involvement of politicians.


*“A lot of people don’t trust politicians”.*
[DŚ-2]

Social media, particularly the Facebook, Tik-Tok, and YouTube profiles of liked persons, were mentioned as one of the possible channels for promoting CRC SP. However, according to the participants, the indicated media have some limitations, such as the difficulty of reaching the elderly or the flood of other information that could interfere with the reception of messages that concern preventive examinations. 

In terms of the content that could be included in information and education campaigns, one participant mentioned the presentation of alarming epidemiological statistics in all possible promotion channels, which would increase public awareness. According to another participant, it is essential to list the benefits of preventive examinations, such as the detection of the disease at an early stage of its development, which enables a complete cure of the disease or a longer life.

*3.* 
*Preferences of the healthcare institution providing the CRC SP*


When asked what conditions should be met by the health centre/clinic to encourage participation in free colonoscopy, participants primarily indicated the broadly understood competencies of a medical staff conducting the examination. In terms of competencies, usually, high medical professional qualifications (hard competencies) were indicated. 


*“I don’t ask the doctor to be an angel and smiler. They should be primarily a good professional”.*
[MA-3]

However, many statements stressed the soft skills of medical staff (communication skills, good contact with patients, empathy). 


*“It’s about the way how the doctor approaches the patient. Even though you know that they have to be professional, you feel more safe if they have a friendly attitude.”*
[MA-2]

The gender or age of the doctor performing the examination did not matter much to participants, although some statements indicated a relationship between age and experience. Several opinions also mentioned the feeling of discomfort when being examined by a member of the opposite sex. Women compared colonoscopy to a visit to a gynaecologist: 


*“I feel uncomfortable when a gynaecologist is a man.”*
[DŚ-1]

The participation of trainees in the examination also causes discomfort: 


*“...it’s a very embarrassing examination, and there are seven students who are watching and saying ‘maybe I’ll try it now, see what happens’. I know they are learning, but not every patient is ready for that.”*
[LU-2]

The issue of the psychological comfort that should be provided to patients during the examination also appeared in other statements, such as the lack of possibility for third parties to enter the office during the examination. 

Issues such as the operating conditions of the centre (standard of premises) are much less important. During the interviews, attention was paid to ensuring medical standards and high-quality medical equipment with which the examination is performed. According to the participants, the centre providing screening colonoscopy services should, first of all, provide proper (optimal) sanitary and infrastructural conditions.

When asked about the possibility of setting a convenient date and time for an appointment at a centre providing screenings, most participants considered this factor to be important. Flexibility as to the choice of day of the week and/or time of day was important, especially for active people. One participant also mentioned the need to prepare appropriately for the screening the day before. This involves taking an extra day off work. 


*“... it is known that before the examination you have to drink this preparation, so a day of holiday is already lost if you are a working person. And if it would be possible to do the examination on Monday, then we can drink it on Sunday, and we don’t lose a day then.”*
[LU-1]

## 4. Discussion

Until now, a very limited number of qualitative studies have been conducted in Poland to identify the expressed expectations and preferences of patients regarding the organizational conditions of CRC screening programme implementation.

In terms of factors that could be reflected in increased patient participation in CRC SP, encouragement from healthcare professionals, especially the primary care physician, is the most frequently mentioned [14,15]. Patients expect to be proactively interviewed by their primary care physician and offered the opportunity to participate in the examinations to which they are entitled (by way of guaranteed health care and individual conditions qualifying them for screenings). Increasing the involvement of the occupational physician doctor in this area was also assessed as a potentially effective factor, while it was also pointed out that there is limited, occasional availability of occupational health services, as well as a potentially lower level of trust compared to GPs.

In addition to the recommendation of physicians to perform screenings, the literature also indicates the need for educational activities on primary and secondary prevention (using appropriate tools [14,15,16] and maximising the use of waiting rooms for this purpose and ensuring adequate resources in the clinic [14]. The results of this study indicate that the HCE could play a supportive role in the medical entities (provided that it is highly qualified and performs its tasks under medical supervision and in cooperation with nurses and clinic administration). Currently, this function in Poland has only been described in the form of theoretical assumptions; therefore, it was assessed in our study as completely new and previously unknown to the participants. This unawareness may be due to limited trust and difficulty in seeing the role of the HCE in promoting screening from the perspective of some participants.

The results of the study indicate that the direct method of reaching the target group for population-based, targeted screening should include sending named letters together with a clear patient leaflet from the primary care clinic [17].

Among other factors presented in the literature, sending a reminder letter [18] and sending immunohistochemical tests to low-income individuals [19] have been reported to be effective in improving the uptake of population-based screening targeted at a specific target group. Within the Polish PICCOLINO Study (a multicentre, randomised trial conducted in 2019–2020 on a sample of 12,485 people), it was shown that offering a combination of the FIT test and colonoscopy as a sequential strategy (when an invitation for the FIT test is sent if there is no response to a named invitation for preventive colonoscopy within 6 weeks) or active choice (when named invitations are sent for both colonoscopy and FIT test with the option for the patient to choose which test they would prefer to attend) increases participation in CRC SP [20].

Effective strategies for increasing participation in CRC screening include ‘navigating’ the patient through the health system and other practices that overcome organizational barriers, including a reminder system and monitoring patient outcomes, and community engagement [21,22].

The literature also indicates the important role of communication between the primary care physician and the patient, shared decision-making regarding preventive screenings [23] and empathy on the part of the medical staff [24]. As part of this study, high soft skills such as communication skills and good contact with patients were rated as important; however, doctors’ professional qualifications (experience) were indicated as one of the key factors that influence willingness to participate in colonoscopy. In light of the results of other studies, such characteristics concerning the health service provider as little experience, implementation of the examination as part of an individual practice, and a female doctor performing the examination were not significant when deciding to participate in a screening colonoscopy [25,26].

The research had two main limitations. Firstly, the way in which the interviews were organized and conducted. The final design of focus groups was characterized by: (1) a non-equivalent number of participants across groups due to variables adopted, (2) the lack of separate groups for those with previous participation in CRC screening and those with no experience of screening) (as transcriptions from the interviews only enabled the identification of statements of particular groups, not individual participants, the study authors could not determine what the differences in opinion would be due to the experience of colonoscopy).

Secondly, due to the conditions of the COVID-19 pandemic, the interviews took place online while maintaining the health and safety of participants and interview moderators. During the FGI, the participants’ faces were not visible – the moderator could only hear the participant’s voice (such conditions may have affected the interview atmosphere).

## 5. Conclusions

The qualitative study allowed us to describe potential factors that may contribute to increased attendance rate and regularity of patient participation for the CRC SP. These include:(1)An increase in participation of primary care clinics in CRC SP promotion and recruitment of participants (assessment of CRC risk and eligibility for the programme that is performed by primary care physicians, provision of information concerning primary and secondary prevention, sending letters or making phone calls, use of time spent by patients in the clinic for educational purposes);(2)The use, in addition to a registered letter, of other forms of direct outreach to those who are encouraged to participate in the CRC SP, such as a phone call made by medical or administrative staff from primary care clinics;(3)Strengthening and improving soft skills in the healthcare workforce in order to create patients’ trust and a positive attitude towards CRC SP;(4)Convenient date and time adjustment for an appointment at a centre providing screenings.

Although the identified factors relate directly to the CRC SP in Poland, the universal nature of the majority of the results of this study means that they can also be taken into account by decision-makers who develop analogous programmes in other countries.

Patients were confirmed to have clear expectations and preferences for the organizational conditions of the CRC SP. Their nature allows them to be treated as one of the potential criteria for selecting critical parameters of SPs (especially for CRC) and, at the same time, a premise for redefining the organizational conditions of the programme towards optimizing its effectiveness.

The far-reaching benefits of incorporating a patient perspective in CRC SP design can only be obtained if the expressed expectations and preferences of patients have a real impact on the choices made by decision-makers.

## Figures and Tables

**Table 1 healthcare-11-00371-t001:** Structure of research questions.

Subject	Research Questions
Preference for actions concerning the promotion of preventive examinations and education aimed directly at the patient	Who should assess individual risk factors for CRC? Who should determine whether the patient is eligible to participate in CRC SP (according to the terms of the CRC SP)? Who should inform patients about the possibility of a screening colonoscopy? (probing questions included issues like occupational health physician, healthcare educator, and any other medical staff) What are the other effective forms of directly reaching and encouraging patients to participate in the CRC SP? (probing questions included issues like prior telephone consultations, SMS messages from SP organizer, anonymous email, or chat with a doctor)
Preferences for actions on the promotion of population-based targeted screening	What should a nationwide, effective screening promotion look like so that healthy people at high risk for CRC come in for screening?
Preferences of the healthcare institution providing the CRC SP	What conditions should a medical centre/clinic fulfil for you to encourage participation in free colonoscopy (CRC SP)? (probing questions included issues like the conditions in which the centre operates, the centre’s reputation, the gender or age of the doctor performing the examination, and ensuring the option of colonoscopy under general anaesthesia).

Source: Author’s own study.

## Data Availability

Not applicable.

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
