# Peer review of "Patients’ Expectations and Preferences for the Organizational Conditions of the Colorectal Cancer Screening Programme in Poland: A Qualitative Analysis"

_healthcare, 2023, doi:10.3390/healthcare11030371_

Round 1

Reviewer 1 Report

Thank you for the opportunity to review this important and interesting study. I have read the paper with pleasure.

Please find below my few remarks and questions regarding the article.

How did you incentivize the participants of the study? Did they receive any compensation for their time?

Table 1 – probe questions or probing questions?

Line 400 – whose?

Have you asked your participants about the form and content of the proposed invitation letters? It seems that not the letter itself but what is in it, is crucial for patient participation in screening programs. There are some studies and good practices (especially from US) on how to make the most of this promotion tool. 

Author Response

Dear Reviewer,

We sincerely thank you for reading the article and preparing the review. We are pleased with the positive opinion. We have made the following changes to the content of the article:

  • we added a sentence regarding the compensation of participants: „Participants were compensated for their participation in the study as part of their membership in the research agency's panel”. The form and value of the compensation are trade secrets.
  • we improved „probing questions” in Table 1.
  • we removed the unnecessary word „whose?” in line 400.
  • unfortunately, our survey did not address the issue, related to the form and content of the proposed invitation letters. As we begin to work on the next stage of the study (quantitative) we will take into account the your rightfully suggestion.

Yours sincerely,

Katarzyna Kędzior

Reviewer 2 Report

The study by Gas et al analyzes the responses from individuals regarding colorectal cancer (CRC) screening programs in Poland. The study uses a qualitative approach wherein the subjects are asked questions during an interview. It is an interesting survey with an aim to increase the screening rates for CRC in Poland. However, the authors recruit a very small sample size from only 5 Polish voivodships, with no explanation why other voivodships were excluded. Additionally the authors have recruited people who have participated in CRC screening programs previously, which could lead to a bias. The survey was not blinded, the responses were based off a discussion with the participants which could ;had to error as they would be authors’ interpretation of their discussion. The authors should also address 

  • if there are other strategies or protocols followed by other countries in EU, which are responsible for higher CRC screening rates in those countries. If education, lifestyle (urban/ rural), or insurance could play a role in the participation of CRC screening. 
  • The source of the questionnaire and data is listed as “Authors' own elaboration “ or “author's own study.” 
  • Define the six focus groups and what was exactly meant by “Sampling continued until saturation was achieved. “ 
  • Running the codes manually could lead to errors. 

Author Response

Dear Reviewer,

We sincerely thank you for reading the article and preparing the review, as well as giving us the opportunity to submit a revised draft of the manuscript. Below are our point-by-point responses to your comments and concerns:

  1. „However, the authors recruit a very small sample size from only 5 Polish voivodships, with no explanation why other voivodships were excluded”. Substantiation: We selected 5 of the 16 voivodships, due to the limitations of the project's budget. The methodology for selecting the voivodeships included: (1) ratios of hospital services provided for colon cancer per 100,000 (varying levels of demand for CRC treatment services (Appendix B.) / line 124); (2) representation of the northern, southern, western, eastern, central parts of Poland (representation of geographically diverse regions of Poland / line 125). The size of the sample was adequate to the research questions and enabled the formulation of final conclusions.
  1. „Additionally the authors have recruited people who have participated in CRC screening programs previously, which could lead to a bias”. Substantiation: People who have participated in CRC screening programs in the past, are asymptomatic of CRC and who had at least one first-degree relative (parents, siblings, children) with CRC in their family, are also the target group of the SP CRC in Poland. Considering the small size of the study group, the answers were not analyzed in terms of differences in the characteristics of the respondents.
  1. „The survey was not blinded, the responses were based off a discussion with the participants which could ;had to error as they would be authors’ interpretation of their discussion”. Substantiation: In qualitative research, compared to quantitative research, statistical inference is not used. The purpose of this research is not to find out how many people think this way but how differently people think. Qualitative research provides an opportunity to determine why, what makes them interested in participating in screening or not interested in participating in screening. Qualitative research also does not use blinding methods like clinical trials. This is only the first, exploratory stage of our research. The detailed categorization and aggregation of responses we made allowed us to avoid individual interpretation by the authors. In addition, the authors of the publication listened to the recordings of the interviews, which allowed us to fully understand the context of the statements and avoid overinterpretation. The researchers who analyzed the statements were not the interviewers. The interview was conducted by a professional research agency.
  1. „The authors should also address if there are other strategies or protocols followed by other countries in EU, which are responsible for higher CRC screening rates in those countries. If education, lifestyle (urban/ rural), or insurance could play a role in the participation of CRC screening”. Substantiation: The study aimed to describe factors that, from the patients' perspective, could increase the attendance rate and regularity of participation for the colorectal cancer screening programme (SP) / line 76-79.  We did not use foreign/UE protocols, as we were only interested in the opinions of Polish patients. Poland has a system of mandatory health insurance, and SPs are fully financed from the budget of the Ministry of Health. Selection criteria for the study population were representative due to sex, area of residence (rural, urban), education (secondary school/highschool/higher education), experience related to participation in screening programme (lack of experience/with experience). Considering the small size of the study group, the answers were not analyzed in terms of differences in the characteristics of the respondents.
  1. „The source of the questionnaire and data is listed as “Authors' own elaboration “ or “author's own study” – Improved.
  2. „Define the six focus groups and what was exactly meant by “Sampling continued until saturation was achieved“ Running the codes manually could lead to errors”. Substantiation: The six focus groups were described in Appendix C. Table 1. Sociodemographic characteristics of participants. Sampling continued until saturation against particular inclusion criteria (e.g., gender: selected participants in equal groups of 12 women, 12 men, etc.). The transcribed interviews were coded independently by two researchers, with cross-checking.

Thank you again for the insightful and valuable comments to our paper.

Yours sincerely,

Katarzyna Kędzior 

Reviewer 3 Report

This is a paper that tries to explore how best to approach asymptomatic people so as to increase the likelihood that they will participate in screening for bowel cancer.

I enjoyed reading the paper. It gave some useful insights into how to explore this particular topic and also into how people (in Poland at least) think about preventative healthcare.

I think the paper could be improved and have the following suggestions:

1 - I think more detail is needed to explain how participants were included in the study i.e. how were they recruited? who recruited them? was 24 the chosen number before recruitment began or was this the maximum that agreed to be recruited? How many people said no to being recruited?

2 - It would be helpful to be told directly how many participants were in the 40-49 group and how many were in the 50- 65 (without me having to count it from the data in Table 1. 13 and 11 respectively). Was this split deliberate and calculated in advance?

3 - who did the interviews? What experience or training did they have?

4 - who analysed the data (the paper mentions 2 people)? What experience and training did they have? Did they analyse the data independently?

5 - two smaller points now

- (i) the word voivodship is not one I am familiar with. I think it might help the reader if the reader knows that this means  'province'. This could be used once e.g.  "voivodship (province) of residence", if the authors were keen to retain it throughout the rest of the paper

(ii) I think the sentence "The respondents were compensated for their participation in the study as part of their membership in the research agency's panel." needs some form of further explanation. How were respondents compensated? Did they know they were going to be compensated?

6 - lastly. I think it would be interesting to know, in the discussion, whether the participants' attitudes towards HCE's were unique to Poland, or whether there was similar data from other countries.

I hope that helps

Author Response

Dear Reviewer, 
First, we would like to say thank you for the useful comments to improve the paper. We have addressed the comments:
Comment 1 - we have added an extensive supplementary sentences in the text. 
Comment 2 - we have added an accurate supplementary sentences in the text. 
The split of 11 and 13 was not deliberate and calculated in advance. 12 and 12 were assumed, as mentioned a paragraph above. Since the equality of the groups by voivodeship (province) of residence, place of residence, gender, as maintained, and the size of the groups by age was similar, the team decided to accept the division of the cohorts. 
Comment 3 - we have added an extensive supplementary sentences in the text. 
Comment 4 - we have added an extensive supplementary sentences in the text. 
Comment 5 - we have corrected in the text - voivodship (province); we have added an explanation of compensation. The conditions for participation in our survey are in line with the generally accepted practice of research agencies worldwide. Initially, we felt that detailed information was not important, but in accordance with your suggestion, we have supplemented the detailed description in the text. 
Comment 6 - we have included the results of all the studies we have found on the analogous subject of research so far in the discussion. 

Once again, thank you for your involvement and all your suggestions. In our opinion, the comments have improved the quality of our paper. 

Yours sincerely,
Katarzyna Kędzior 
